# Deep Learning Model for Computer-Aided Diagnosis of Urolithiasis Detection from Kidney–Ureter–Bladder Images

**DOI:** 10.3390/bioengineering9120811

**Published:** 2022-12-16

**Authors:** Yi-Yang Liu, Zih-Hao Huang, Ko-Wei Huang

**Affiliations:** 1Department of Electrical Engineering, National Kaohsiung University of Science and Technology, Kaohsiung City 80778, Taiwan; 2Department of Urology, Kaohsiung Chang Gung Memorial Hospital and Chang Gung University College of Medicine, Kaohsiung City 83301, Taiwan

**Keywords:** computer-aided diagnosis, kidney–ureter–bladder, kidney stone, deep learning, residual network

## Abstract

Kidney–ureter–bladder (KUB) imaging is a radiological examination with a low cost, low radiation, and convenience. Although emergency room clinicians can arrange KUB images easily as a first-line examination for patients with suspicious urolithiasis, interpreting the KUB images correctly is difficult for inexperienced clinicians. Obtaining a formal radiology report immediately after a KUB imaging examination can also be challenging. Recently, artificial-intelligence-based computer-aided diagnosis (CAD) systems have been developed to help clinicians who are not experts make correct diagnoses for further treatment more effectively. Therefore, in this study, we proposed a CAD system for KUB imaging based on a deep learning model designed to help first-line emergency room clinicians diagnose urolithiasis accurately. A total of 355 KUB images were retrospectively collected from 104 patients who were diagnosed with urolithiasis at Kaohsiung Chang Gung Memorial Hospital. Then, we trained a deep learning model with a ResNet architecture to classify KUB images in terms of the presence or absence of kidney stones with this dataset of pre-processed images. Finally, we tuned the parameters and tested the model experimentally. The results show that the accuracy, sensitivity, specificity, and F1-measure of the model were 0.977, 0.953, 1, and 0.976 on the validation set and 0.982, 0.964, 1, and 0.982 on the testing set, respectively. Moreover, the results demonstrate that the proposed model performed well compared to the existing CNN-based methods and was able to detect urolithiasis in KUB images successfully. We expect the proposed approach to help emergency room clinicians make accurate diagnoses and reduce unnecessary radiation exposure from computed tomography (CT) scans, along with the associated medical costs.

## 1. Introduction

Studies based on data from seven countries (Italy, Germany, Scotland, Spain, Sweden, Japan, and the United States) have shown that the prevalence and incidence of kidney stones have been increasing globally [1,2,3]. To address this problem, various methods have been developed to detect and treat kidney stones [4,5]. Computed tomography (CT) is a particularly accurate diagnostic method, with sensitivity and specificity ranging from 94% to 100% and 92% to 94.2% for kidney stones, respectively [6,7]. Therefore, CT is the gold standard for kidney stone diagnosis. However, CT is costly and requires a higher radiation dose than plain film X-ray imaging. For example, the radiation dose of an abdominal CT scan ranges from 8 to 34 mGy [8,9], in contrast to the lower dose of 2.47 mGy required to record a kidney–ureter–bladder (KUB) image [10]. Similarly, a stomach CT requires 50 times the radiation dose of a plain film stomach X-ray [11]. Although low-dose CT reduces the radiation dose from 25 to 17 mGy for abdominal CT scans, this value is still higher than that of plain film X-ray imaging [12]. Therefore, plain film X-ray imaging may be considered as a cost-effective alternative to CT, which also causes less harm to the body. However, X-ray images also have a propensity for false positives owing to their 2D nature, and their resolution does not suffice to identify abnormalities in dense tissues [13]. For example, KUB images have sensitivities of only 44–77% and specificities of 80–87% in kidney stone detection [14], which are considerably inferior values compared to those of CT. Improving the sensitivity of KUB imaging for use with kidney stones could therefore allow Χ-ray scans to be used as a widely applicable option to diagnose the condition, which would also reduce medical costs.

Medical imaging has become increasingly important in clinical diagnosis. X-rays, magnetic resonance imaging (MRI), and CT are among the most common medical imaging modalities. Medical image processing often depends on the experience of radiologists, who must analyze these images and draw conclusions using a subjective approach. As the variety and number of medical imagery techniques have significantly increased in recent years, manual analysis has become increasingly time-consuming and labor-intensive. To address this problem, machine learning models have been used to replicate human visual perception mechanisms to enable computational systems to automatically classify medical images as diagnostic aids. Computational techniques have become significantly more powerful over the last several decades owing to rapid advances in AI and computing hardware, and the use of computer-aided methods to analyze and process medical imagery has become incredibly useful for diagnosticians both in classifying and in augmenting those images. Thus, computer-aided diagnosis (CAD) has become theoretically and practically significant as an important trend in medical science. The use of computer vision to automatically analyze and process medical images has several unique advantages [15,16,17]. For example, this approach leverages the immense computing power of modern hardware to achieve rapid and accurate analysis and processing, which renders its findings immune to fatigue or cognitive issues with information overload. Furthermore, computer technologies and networks can enable the rapid transfer of clinical data to facilitate the rapid and accurate diagnosis of patients in remote locations. Machine learning algorithms designed to diagnose and detect various medical conditions have become a topic of active research, and the accuracy of artificial intelligence classifiers used to predict various related data of patients with kidney stones has also increased [18]. Since the emergence of the first convolutional neural network (CNN) (LeNet-5 [19]) in 1989, CNN models have continued to improve, and deep CNNs have been shown to perform extremely well in medical image processing [20,21,22,23]. The accuracy of CAD methods has also benefited from the progressive improvement of such models [24,25]. In urology, several studies have considered the use of neural networks to aid in the diagnosis of urinary diseases based on CT imaging [26,27,28]. The application of CAD to X-ray examinations has also yielded impressive results. For example, a CNN model trained to diagnose urinary tract stones from plain film X-ray imaging using pre-processed images showed a sensitivity of 89.6% and PPV of 56.9% for kidney stones [29].

Thus far, KUB imaging is still considered as a first-line examination for urolithiasis detection in the emergency room due to its convenience, low cost, and low radiation dose. However, only highly experienced urologists or radiologists can diagnose urolithiasis correctly from KUB images. Furthermore, emergency physicians who arrange KUB images cannot immediately obtain a formal report from the experts. Hence, emergency physicians without the necessary specialized experience are highly likely to either make incorrect diagnoses or choose to arrange non-contrast CT scans for such patients, which may delay further treatment or increase medical costs and the radiation dose. In this study, to address this challenge, we constructed a CAD system based on a deep learning model trained to help emergency physicians make correct diagnoses of urolithiasis from KUB images.

## 2. Materials and Methods

### 2.1. Datasets

The protocol of the present study was approved by the Institutional Review Board of Kaohsiung Chang Gung Memorial Hospital. A total of 355 KUB images were retrospectively collected from 104 patients from Kaohsiung Chang Gung Memorial Hospital who were diagnosed with stones in their upper urinary tract. The presence of stones in the upper urinary tract shown in these 355 images was formally reported by radiologists and then confirmed on a case-by-case basis by two experienced urologists specializing in urolithiasis. The set of KUB images was first divided into groups of training images with single or multiple urinary tract stones, and the dataset was augmented through various image pre-processing operations to produce a total of 1130 images. Then, these 1130 images were divided into three datasets, with 856 images used to train the network, with 80% (684 images) allocated to the training process itself and 20% (172 images) for validation. The remaining 274 images were used to evaluate the performance of the trained model based on several metrics and to test its generalizability. A flowchart of the work performed in this study is shown in Figure 1.

### 2.2. Image Pre-Processing

First, a Mask R-CNN model was trained to detect the spine and pelvis bones in the KUB images [30,31], and the trained model was applied to mask most of the high-brightness regions. The images were then centered on the spine, and the area above the pelvis region was segmented. Because identifying abnormalities in highly dense tissues using plain film X-ray images is difficult, we aimed to exclude factors that tend to lead to the misidentification of features around the kidneys [13]. Furthermore, because of the characteristics of plain film X-ray imaging, dense tissues appear with higher brightness. Hence, histogram equalization [32] can easily lead to overexposure of the image and thus influence the detection of urinary tract stones. The effects of masking on an X-ray image may be observed from a histogram. Contrast-limited adaptive histogram equalization (CLAHE) has been used to enhance contrast in KUB imaging [33], which allows stones to be distinguished from the background through their brightness and also prevents overexposure from excessively high brightness. In this study, we compared the effects of histogram equalization and CLAHE on the KUB images. Finally, patches with a size of 100 × 100 pixels were cropped from the pre-processed X-ray plain films. Patches containing a stone were cropped with the stone at the center, whereas patches without a stone were randomly cropped from the pre-processed plain film X-ray images [29].

### 2.3. Data Augmentation

Numerous studies have shown that data augmentation is effective in preventing overfitting, which is more likely to occur for CNN models trained with smaller datasets [34,35,36,37]. In particular, relatively large datasets of medical images for analysis are often difficult to obtain. Furthermore, the generalizability of learning models depends on the diversity of the data samples [38,39,40]; the more generalizable a model is, the more accurate its results with images that were not present in the training dataset. During training, 100% accuracy can be achieved very quickly, although the prediction accuracy of a model trained in this way is typically reduced. To prevent overfitting and improve sample diversity, we performed data augmentation prior to the training process by rotating, vertically and horizontally translating, magnifying/shrinking, and shear-mapping the original images. However, in contrast to conventional data augmentation methods, the images in the training set were randomly augmented after each iteration of the training process to produce a dynamically augmented dataset. This approach also greatly reduces memory consumption.

### 2.4. Deep Learning Models

We adopted a ResNet-50 architecture as the CNN model in this study. Many studies have shown that the fineness of detail that can be extracted by a CNN increases with the depth of the network. However, He et al. (2016) demonstrated that performance degrades if the depth increases beyond a certain point [41]. Residual network (ResNet) architectures are based on residual blocks comprising convolutional, activation, and batch normalization (BN) layers F(x), and a shortcut connection that reproduces the input x. Because the output of a residual block is H(x) = F(x) + x, the layers in a traditional network effectively learn the difference between the true output and x, i.e., the residual, as shown in Figure 2. Therefore, for the simple case in which the network has not learned any features and the input is already optimal, F(x) is approximately 0, or H(x) = x (i.e., the identity relation). This solves the degradation problem and allows for extremely deep networks. The ResNet architecture is shown in Figure 3. By employing deep learning models for image classification, images can be automatically classified and labeled for various applications [42].

### 2.5. Technical Details and Evaluation Metrics

The data were divided into training, validation, and testing sets. The validation set was taken from the training set at a 20:80 ratio. The testing set consisted of 24% of the total data and was used to evaluate the accuracy of the trained ResNet model. The inputs of the image classification model consisted of images with a size of 224 × 224 px, and the urinary tract stone images were diversified using data augmentation techniques (random rotation, horizontal/vertical translation, magnification/zooming out, and shear-mapping). The Keras API with the Tensorflow platform (version 2.9.1) was used to construct the ResNet model. Ranger [43], which was created by combining RAdam [44] with LookAhead [45], was used as an optimizer. As with Adam, RAdam converges quickly and achieves a level of optimality similar to that of SGD. Furthermore, RAdam converges similarly with different learning rates, whereas Adam and SGD are much more sensitive to the learning rate and require optimization. Binary cross-entropy was used as a loss function. The predictions were used to construct a confusion matrix of four possible outcomes (see Figure 4). Correct predictions are either true positive (TP) or true negative (TN), whereas incorrect predictions are either false positive (FP) or false negative (FN). These outcomes were used to construct seven metrics to evaluate the performance of the model, including accuracy, sensitivity, specificity, precision, F1-measure, the receiver operator characteristic (ROC) curve, and the area under the ROC curve (AUC). Accuracy is defined as
(1)$$\mathrm{accuracy}=\frac{\mathrm{TP}+\mathrm{TN}}{\mathrm{TP}+\mathrm{TN}+\mathrm{FP}+\mathrm{FN}}$$

Although this is a simple metric, accuracy is susceptible to bias for unbalanced training data. Therefore, we also used the four other metrics mentioned above. Sensitivity, also known as recall, provides the proportion of patients with kidney stones who were correctly predicted as having the condition. Sensitivity is given by
(2)$$\mathrm{sensitivity}=\frac{\mathrm{TP}}{\mathrm{TP}+\mathrm{FN}}$$

Specificity provides the proportion of patients without kidney stones who were correctly predicted as negative for the condition, and is given by
(3)$$\mathrm{specificity}=\frac{\mathrm{TN}}{\mathrm{FP}+\mathrm{TN}}$$

Precision is the proportion of patients who actually had kidney stones among all persons predicted to have the condition, and is given by
(4)$$\mathrm{precision}=\frac{\mathrm{TP}}{\mathrm{TP}+\mathrm{FP}}$$

The F-measure is a comprehensive measure of performance, of which F1-measure is a special case. If *β* is equal to 1, the F1-measure will equally reflect both recall and precision. However, if *β* is greater than 1, the F1-measure depends more on the recall than the precision, and vice versa. The F1-measure is given by Equation (5); higher values indicate better performance.
(5)$$F_{\beta }-\mathrm{measure}=\left(1+\beta^{2}\right)\frac{Precision\times Recall}{\left(\beta^{2}\times Precision\right)+Recall}$$

The last two metrics are the ROC curve and AUC. The ROC curve is obtained by plotting the true positive rate (TPR) (y-axis) against the false positive rate (FPR) (x-axis). The TPR is the proportion of positive predictions that are actually positive, and the FPR is the proportion of positive predictions that are actually negative. Therefore, the ROC curve represents the relationship between the FPR and TPR in the model. Because the (0, 1) point corresponds to a perfect classification, the performance of a model is proportional to the closeness of its ROC curve to the top-left corner. Similarly, the AUC is the area under the ROC curve, which increases as the ROC curve approaches the top-left corner. Hence, the performance of a classification model is directly proportional to the AUC.

## 3. Results

### 3.1. Image Pre-Processing with Histogram

When histogram equalization (HE) is performed on KUB images, overexposure often occurs at the spine and pelvis, which tends to affect training negatively. Figure 5 shows an HE-processed image overexposed around dense tissues (bone), especially around the pelvis and spine, which may induce deviations during the feature extraction process. Therefore, all high-density regions in the KUB images must be masked. As shown in Figure 6, masking the spine and pelvis greatly reduced the high-intensity area of the images. Nonetheless, some overexposure still occurred at the rib cage, which is a common problem in HE. To prevent image overexposure from HE, CLAHE was performed on the KUB images. From Figure 7, it may be clearly observed that the CLAHE-processed image exhibits relatively little overexposure. Therefore, the CLAHE-processed KUB images were considered suitable for the observation of kidney stones.

### 3.2. Effects of Data Augmentation on Training

We trained the ResNet model using both augmented and non-augmented datasets. Data augmentation was performed by rotating, horizontally and vertically translating, magnifying/shrinking, and shear-mapping the original images. In the augmented dataset (which contained the same number of images as the non-augmented dataset), these data augmentation procedures were randomly applied to every image after each iteration to ensure that the training data differed between iterations. The results were then compared in terms of accuracy and loss. Figure 8a,b show the results obtained with and without data augmentation, respectively. Although the accuracy increased much more rapidly when the model was trained on the non-augmented dataset, it was unable to obtain a similar level of accuracy on the validation dataset in that case.

### 3.3. Experimental Results

The model was trained for 50 epochs with an initial learning rate of 10^−5^. Because appropriate decreases to the learning rate are conducive for optimization, the learning rate was multiplied by 0.5 if the validation loss was not updated for five continuous epochs. The epoch-wise changes in accuracy and loss are shown in Figure 9a,b, respectively. It can be observed that the process of training from 0 epochs to 20 epochs converged rapidly. The accuracy and loss of the training set and the verification set were close, indicating that the model learned features in the initial stage well and classified them accurately. The subsequent loss from the 20th to the 50th epochs gradually converged to the optimal solution as the training ended. According to the confusion matrix shown in Table 1, the final accuracy of the model was 0.977, and its accuracy on the testing set was 0.982. Sensitivity is the ratio of patients with kidney stones who were correctly identified as positive cases, while precision is the ratio of correct diagnoses among positive cases. Therefore, a high sensitivity implies that false negatives are rare. Specificity is the ratio of patients without kidney stones who were correctly diagnosed as negative cases. Therefore, a model with a high specificity is unlikely to misdiagnose healthy subjects as positive cases. The F1-measure is the harmonic mean of recall (sensitivity) and precision, which summarizes the performance of a model. In kidney stone classification, the focus is on sensitivity, as the primary goal is to correctly identify patients who suffer from kidney stones. The sensitivity, specificity, precision, and F1-measure scores of our model were 0.953, 1, 1, and 0.976 on the validation set and 0.964, 1, 1, and 0.982 on the testing set, respectively (see Table 2). The ROC curves were also plotted to test the effectiveness of the model, and their AUCs were 0.995 and 1 on the validation and testing sets, respectively (Figure 10a,b). When AUC > 0.5, the classification performance of a classifier is better than random guessing, and the model has positive predictive value. The AUC value of our model was quite close to 1, which shows that the performance of our model was close to that of a theoretically perfect classifier, and it was effective in predicting positive samples correctly.

### 3.4. Comparison of Accuracy with an Existing Method

The sensitivity, precision, and F1-measure of our method were 0.964, 1, and 0.982 on the testing set, respectively. Another CNN-based deep learning model [29] trained to detect kidney stones in pre-processed plain film X-ray images was also used for comparison, and the sensitivity, precision, and F1-measure of this model were 0.985, 0.762, and 0.862, respectively, as shown in Table 3. Therefore, the performance of the proposed method was superior. This improvement may be attributed to the following factors. In addition to the differences in data collection, we utilized iterative data augmentation and various image pre-processing techniques. Data augmentation is commonly used in studies on medical imaging, especially to address overfitting with small datasets, and has achieved excellent results [46,47,48]. The use of CLAHE instead of HE for image pre-processing also helped reduce overexposure of the plain film X-ray images.

## 4. Discussion

In this study, we trained a CNN model to classify KUB images according to the presence of kidney stones. Although few studies have been conducted on the use of plain film X-ray images to detect kidney stones, the results are promising. According to a recent systematic review of recent AI advancements in urology by Dai et al. [49], only a single study used KUB images [29]. Other studies largely considered machine and deep learning models based on CT images, such as a work by Parakh et al. [50]. First, the advantages of plain film X-ray images include their low dose and cost, which enables them to be used in a wide range of medical institutions. Second, many deep learning models cannot accurately detect small objects or features, and kidney stones usually occupy an extremely small number of pixels in a KUB [51]. To address this problem, the images were cropped to enlarge the size of kidney stones and to train the model more easily. Third, the accuracy and generalizability of the model can be further improved by increasing the size of the training dataset. In the context of medical imaging, some plain film X-rays of kidney stones exhibit rarely-encountered patterns and features, which can make determining whether a kidney stone is present difficult. However, owing to their rarity, learning models cannot be trained on such images. By contrast, in most plain film X-ray images used to train the model, kidney stone(s) could be observed with the naked eye. If a large number of plain film X-ray images with difficult-to-observe kidney stones could be collected, the generalizability of the model could then be enhanced with further training to produce a highly reliable CAD tool. Although the kidney stones that our model was able to detect were obvious in the X-ray images, the model was nonetheless able to differentiate plain film X-ray images according to the presence or absence of such stones, which demonstrates that this approach can be extended to object detection and segmentation in the future. In deep learning studies on breast X-rays, over 4000 images have commonly been used to train deep learning models [52,53,54,55]. In this study, only 1130 images were used, and the small size of the dataset could have resulted in a poor training outcome. We therefore used data augmentation techniques to avoid severe overfitting and achieve adequate generalizability. In this study, we only used conventional data augmentation techniques as noted above rather than a generative adversarial network (GAN). Because GAN models have been successfully used to generate medical images [38,39,56,57], this approach remains as a potential direction for future research.

## 5. Conclusions

In this study, we trained a ResNet model to classify KUB images based on the presence or absence of kidney stones. The proposed model presents excellent classification performance in terms of several metrics and can be used in the immediate diagnosis of kidney stones from plain film X-ray images. We draw the following conclusions from the results. (1) The retention of the spine and pelvis bones during image pre-processing exhibited an outsized impact on the accuracy of the model. (2) Overexposure from histogram equalization reduced the accuracy and other evaluation metrics. This can be alleviated through masking and contrast-limited adaptive histogram equalization, which increase the training accuracy and improve the performance of the model. (3) Overfitting can be reduced for small datasets by augmenting the data used to train the model. This process also improves the generalizability of the model on unknown data, which explains why our model performed similarly on the validation and test sets. The proposed approach is expected to reduce the consumption of medical resources and limit the patients’ radiation exposure, which is beneficial for both patients and physicians.

In the future, the proposed ResNet model could be combined with object detection or image segmentation strategies, such as SSD, Inception, or U-Net, to effectively detect very small kidney stones. In addition, we plan to consider topics beyond image classification. Once the classification model is more complete, we plan to study object detection and segmentation methods to locate and label any kidney stone appearing in KUB images, where each image may contain one or many objects of varying types. For object detection, we expect to adopt RetinaNet [58], which adds a single-shot multibox detector (SSD) to the frontend of ResNet and utilizes a focal loss function to improve image classification accuracy on unbalanced data, which is often the case for medical data. However, object detection methods only provide a rectangular bounding box enclosing a feature rather than the exact profile of an object, which can be crucial to diagnose a condition. Therefore, image segmentation is a quintessential part of an AI-driven CAD. To this end, we expect to use CaraNet as an image segmentation model [59]. In a 1000 × 1000 px KUB image, a kidney stone may occupy a region smaller than 20 × 20 px. Because CaraNet is specifically designed for the segmentation of small objects, we plan to study the feasibility of using CaraNet to improve the segmentation of small kidney stones in KUB images in future work.

## Figures and Tables

**Figure 1 bioengineering-09-00811-f001:**
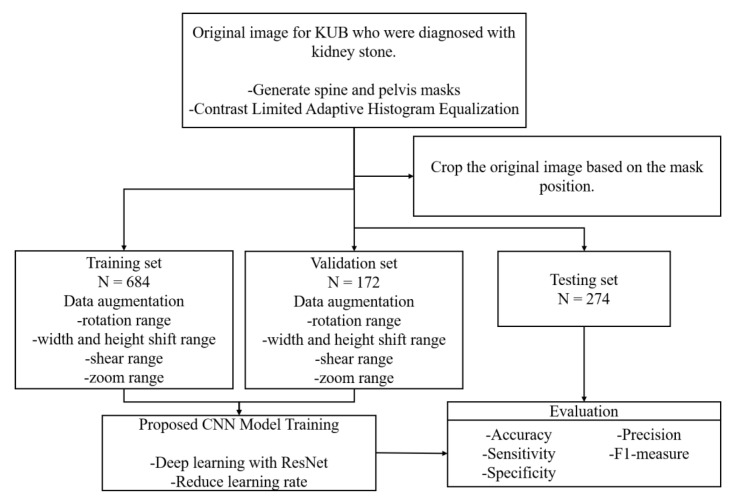
Process flow used in this study.

**Figure 2 bioengineering-09-00811-f002:**
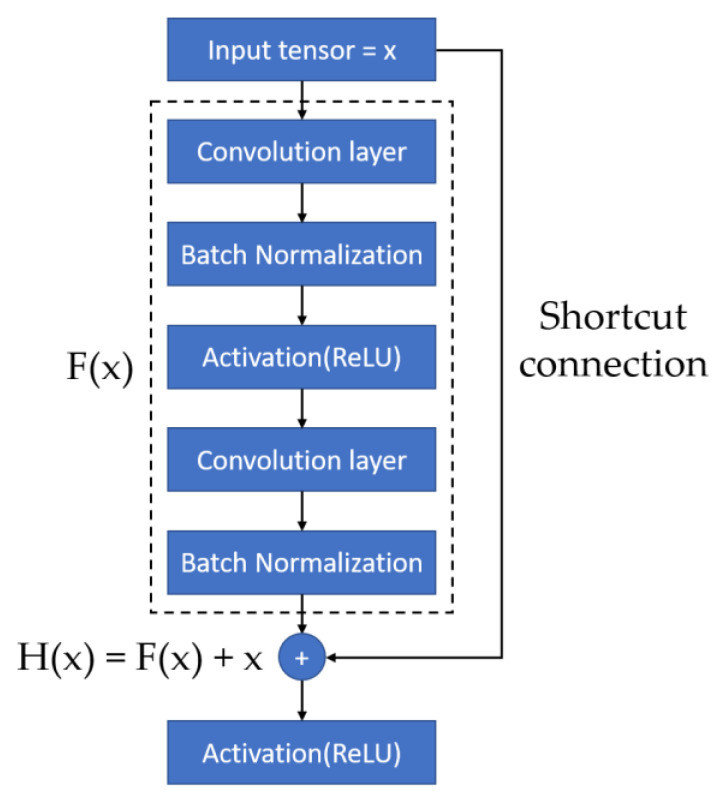
Residual Block [41].

**Figure 3 bioengineering-09-00811-f003:**
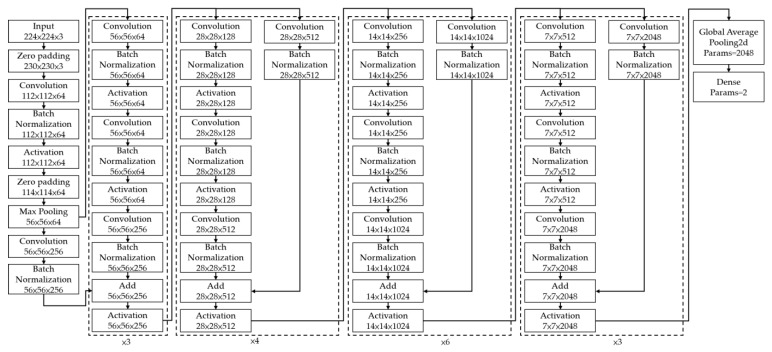
Residual Network Architecture [41].

**Figure 4 bioengineering-09-00811-f004:**
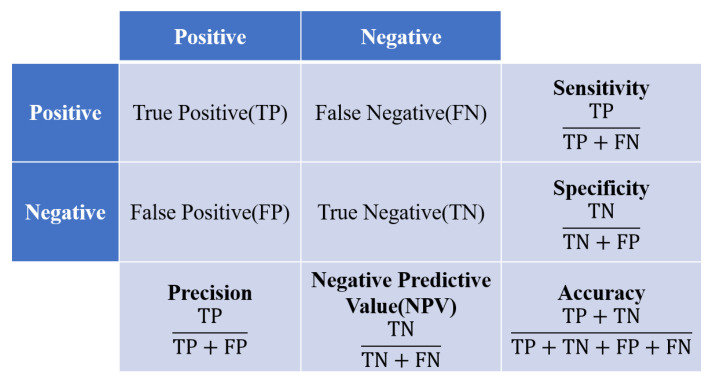
Confusion matrix and evaluation metrics.

**Figure 5 bioengineering-09-00811-f005:**
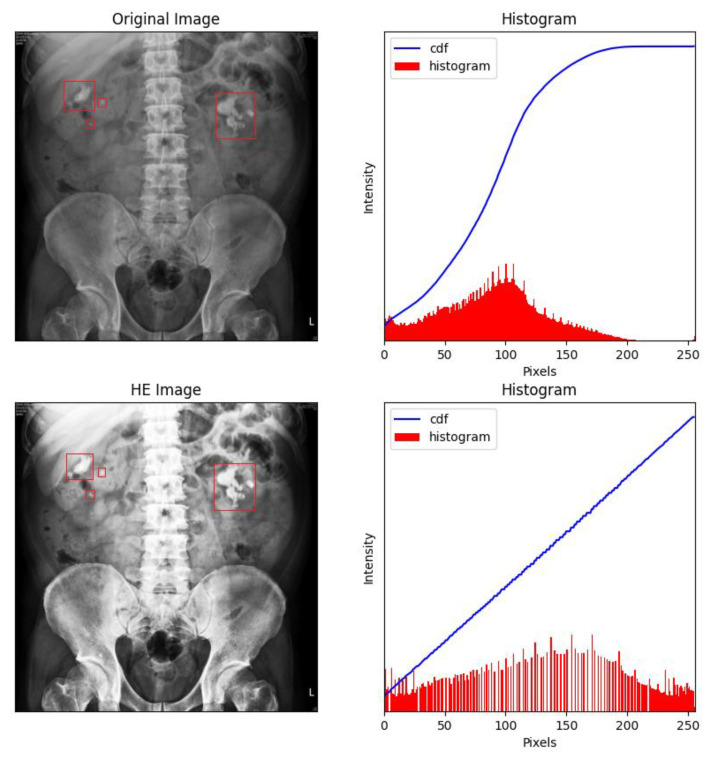
Many areas were overexposed in the HE-processed images, especially around dense tissue such as bone. The renal stones were labelled with red frames by the experts.

**Figure 6 bioengineering-09-00811-f006:**
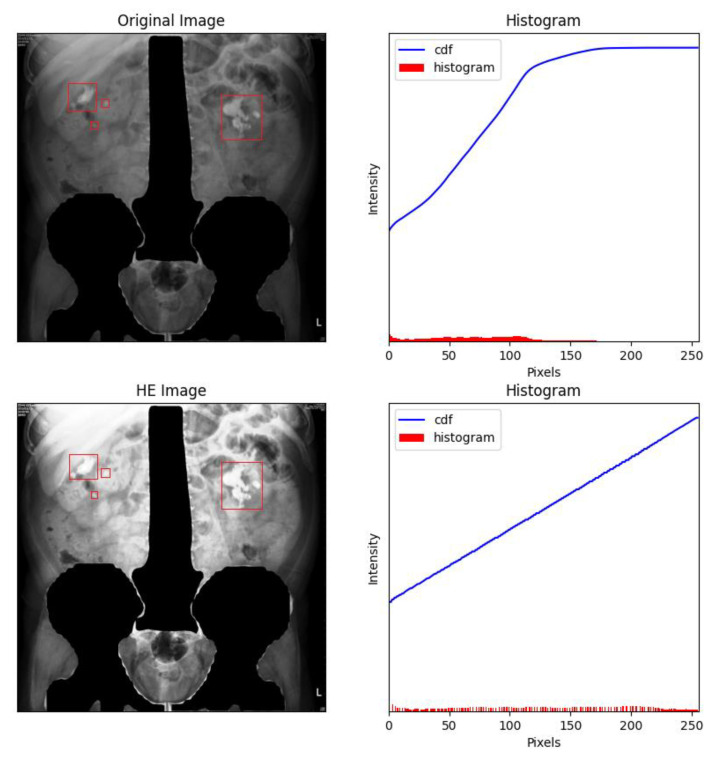
Masking the spine and pelvis greatly decreased the high-intensity areas of each image. The renal stones were labelled with red frames by the experts.

**Figure 7 bioengineering-09-00811-f007:**
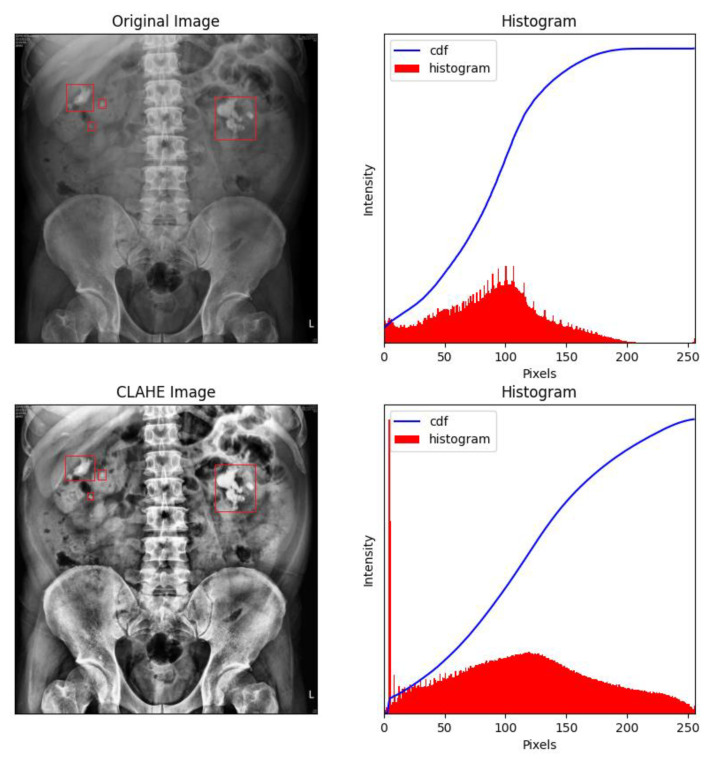
Contrast-limited adaptive histogram equalization (CLAHE) of a KUB image. It can clearly be observed that CLAHE greatly reduced overexposure around the rib cage, which makes identifying kidney stones relatively straightforward. The renal stones were labelled with red frames by the experts.

**Figure 8 bioengineering-09-00811-f008:**
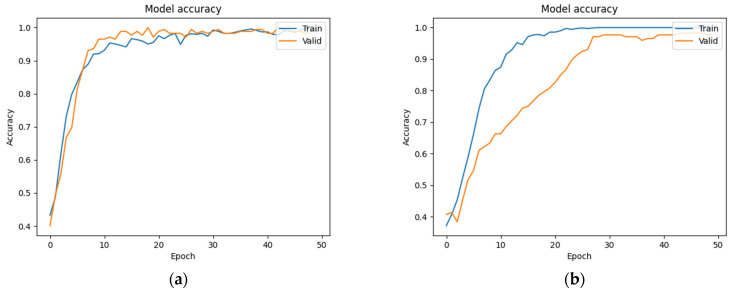
(**a**) Training accuracy using data augmentation. (**b**) Training accuracy without data augmentation.

**Figure 9 bioengineering-09-00811-f009:**
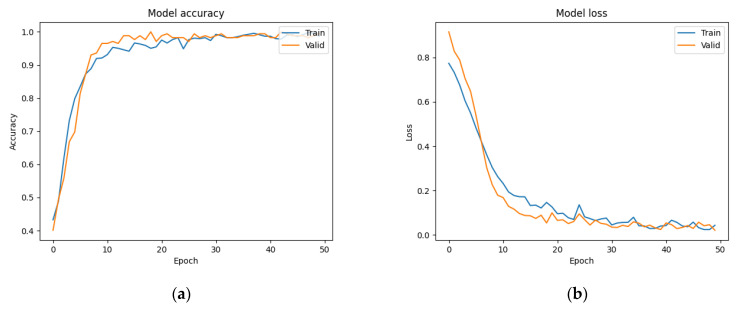
(**a**) Training accuracy and (**b**) training loss of ResNet model on our dataset.

**Figure 10 bioengineering-09-00811-f010:**
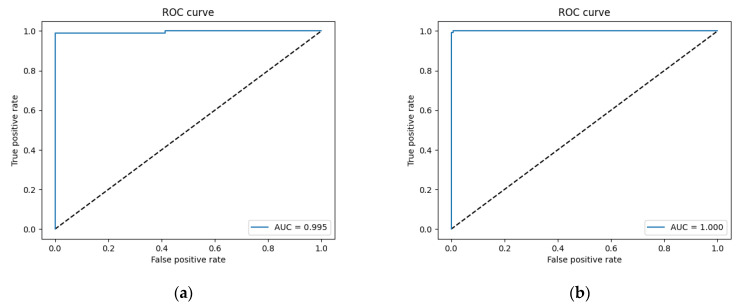
ROC curve of (**a**) validation set and (**b**) testing set.

**Table 1 bioengineering-09-00811-t001:** Confusion matrix of the model.

	TP	FP	TN	FN
Validation dataset	81	4	87	0
Testing dataset	132	5	137	0

**Table 2 bioengineering-09-00811-t002:** Performance of ResNet model on our dataset.

	Accuracy	Sensitivity	Specificity	Precision	F1-Measure	AUC
Validation dataset	0.977	0.953	1.000	1.000	0.976	0.995
Testing dataset	0.982	0.964	1.000	1.000	0.982	1.000

**Table 3 bioengineering-09-00811-t003:** Overall performance with CNN-based model [29].

	Sensitivity	Precision	F1-Measure
Proposed Model	0.964	1.000	0.982
CNN-based model [29]	0.985	0.767	0.862

## Data Availability

The data presented in this study are available on request from the corresponding author.

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
