# Peer review of "Deep Learning Model for Computer-Aided Diagnosis of Urolithiasis Detection from Kidney–Ureter–Bladder Images"

_bioengineering, 2022, doi:10.3390/bioengineering9120811_

Round 1
Reviewer 1 Report
The number of medical images has in-creased dramatically in recent years. Artificial intelligence-based computer aided diagnosis may therefore be an effective way to reduce such fatigue. Herein, in this article a deep learning model for potentially improving the accuracy in classifying kidney ureter bladder (KUB) images in terms of the pres-ence or absence of kidney stones was presented.
The paper is well structured. And can be accepted. Nevertheless, please
1. Make Fig. 1 and Fig. 4-6 bigger.
2. As a broad audience is expected to read the paper, please show on your images (Fig 4-6), the presence of kidney urinary stone. So, it would be more informative to the reader.
3. Please explain in Discussion section, how to make a DNN work better without making bigger datasets. Is there any alternative methods to improve your results?
Author Response
Reviewer 1
The number of medical images has in-creased dramatically in recent years. Artificial intelligence-based computer aided diagnosis may therefore be an effective way to reduce such fatigue. Herein, in this article a deep learning model for potentially improving the accuracy in classifying kidney ureter bladder (KUB) images in terms of the pres-ence or absence of kidney stones was presented.
The paper is well structured. And can be accepted. Nevertheless, please
Reviewer Comment P 1.1—Make Fig. 1 and Fig. 4-6 bigger.
Response: We thank the reviewer for this pertinent suggestion. Following your suggestion, we have reorganized Figure 1. We thank the reviewer for this pertinent suggestion. Following your suggestion, we have reorganized Figure 1 and zoomed out Figures 4-6 (as shown in comments 1.2).
Figure 1. Flowchart of the study
Reviewer Comment P 1.2—As a broad audience is expected to read the paper, please show on your images (Fig 4-6), the presence of kidney urinary stone. So, it would be more informative to the reader.
Response: We thank the reviewer for this pertinent suggestion. Following your suggestion, we have marked the kidney urinary stone in Figures 4–6 as follows:
Figure 4. Many areas are overexposed in the HE-processed image, especially around dense tissue, like bones.
Figure 5. Masking the spine and pelvis greatly decreases high-intensity areas.
Figure 6. Contrast limited adaptive histogram equalization (CLAHE) of a KUB image. It is clear that CLAHE greatly reduces overexposure around the rib cage, which makes it easy to observe kidney stones.
Reviewer Comment P 1.3—Please explain in Discussion section, how to make a DNN work better without making bigger datasets. Is there any alternative methods to improve your results?
Response: Thank you very much for pointing out this shortcoming. Following your suggestions, we have given a brief explanation as follows:
The problem with the neural network is that it requires a large amount of data for training. It is relatively difficult to train small data sets. It is challenging to achieve training on small data sets simply by adjusting the parameters of the neural network and changing the structure. Therefore, it is necessary to supplement data augmentation. Technology allows small data sets to achieve the results of neural network training, and data augmentation is also mentioned in our paper. Different image processing techniques are used to acquire image changes so that the model can be trained to obtain better accuracy.

Reviewer 2 Report
• The authors could better explain how “Related works” is actually related to the current study. It is not clear to the reader how the manuscript is similar to or differs from these related works.
• How did the authors apply the Augmentation technique?
• Revise the abstract section, by highlighting the proposed work and its achievement, instead of mentioning about the datasets.
• Provide high quality figures in the entire manuscript.
• Some recent works about Image Classification Algorithms should be added such that: https://doi.org/10.32604/cmc.2022.019790
• Mention the future scope or your plan to extend this work.
• The authors must show how the effectiveness of the used model be measured
· Check typos and grammar errors with some standard tool, it seems before submission revision, it is necessary to scan, and it will good English checker.
· The contribution of the paper can be neatly elaborated by including the novelty of the paper and the major difference from available models.
Author Response
Reviewer 2
Reviewer Comment P 2.1—The authors could better explain how “Related works” is actually related to the current study. It is not clear to the reader how the manuscript is similar to or differs from these related works.
Response:
We thank the reviewer for this pertinent suggestion. Following your suggestion, we have revised the Introduction section as follows:
Generally, experienced medical experts are needed to diagnose medical conditions from medical images. The manual effort this requires often makes it difficult to obtain timely diagnoses. Therefore, AI-based computer aided diagnosis (CAD) could be used by non-experts to rapidly obtain an accurate diagnosis for subsequent treatment. It is well-known that kidney, ureter, and urinary bladder (KUB) X-ray images only require low radiation doses and are relatively easy to obtain whilst being cost efficient. In addition, KUB imaging can be arranged at any time as a first-line imaging modality for kidney stones. However, this approach has poor diagnostic rates, as a very experienced urologist or radiologist is required to diagnose kidney stones from KUB images. Furthermore, the emergency physician who first saw the patient will not be able to immediately prepare a formal report for the diagnostician. In the absence of this report and sufficient experience, the diagnostician is likely to make mistakes or call for a CT, which delays treatment and increases treatment costs. To address this problem, we constructed a KUB-based deep learning CAD model to help frontline emergency physicians diagnose kidney stones and refer patients to urologists
Reviewer Comment P 2.2—How did the authors apply the Augmentation technique?
Response:
Thank you very much for pointing out this shortcoming. Following your suggestion, we have revised the section 3.2 Effect of data augmentation on model training as follows:
Here, ResNet training was conducted using augmented and non-augmented datasets. Data augmentation was performed by rotating, translating (vertically and horizontally), magnifying/shrinking, and shear mapping the original images. In the augmented dataset (which contains the same number of images as the non-augmented dataset), the aforementioned data augmentation procedures were randomly applied to every image after each iteration. This ensures that the training data are different between iterations.
Reviewer Comment P 2.3—Revise the abstract section, by highlighting the proposed work and its achievement, instead of mentioning the datasets.
Response: We thank the reviewer for this pertinent suggestion. Following your suggestion, we have reorganized the entire abstract section as follows:
The kidney ureter bladder (KUB) image is a radiological examination with low-cost, low-radiation and convenience. It is easy for clinicians in the emergency room to arrange KUB images as first-line examination for the patients with suspicious urolithiasis. However, it is hard to interpret the KUB images correctly by inexperienced clinicians. Also, it is difficult to get a formal report by the radiologist immediately after KUB image examination. Recently, artificial intelligence-based computer aided diagnosis (CAD) can effectively help non-expert clinicians to quickly make a correct diagnosis for further treatment. Therefore, the purpose of this study is to design a deep learning model for CAD of KUB images to help first-line clinicians in the emergency room make accurate diagnosis of urolithiasis. In this study, a total of 355 KUB images were retrospectively collected from 104 patients in Kaohsiung Chang Gung Memorial Hospital who were diagnosed with stones in their upper urinary tract. Then a deep learning model for potentially improving the accuracy in classifying KUB images in terms of the presence or absence of kidney stones was trained using ResNet on pre-processed KUB images and the parameters were tuned and tested. The experimental results indicate that the accuracy, sensitivity, specificity, and F1-measure of the model were 0.977, 0.953, 1, and 0.976 on the validation set, and 0.982, 0.964, 1, and 0.982 on the testing set, in addition, the results also demonstrate that the proposed model competes well against existing CNN-based model. In conclusion, we used a deep learning model to establish a computer-aided diagnosis for urolithiasis detection in Kidney Ureter Bladder images successfully. The model helps clinicians in the emergency room make accurate diagnosis and reduce unnecessary computed tomography (CT) related radiation exposure and waste of medical cost.
Reviewer Comment P 2.4—Provide high quality figures in the entire manuscript.
Response: We thank the reviewer for this pertinent suggestion. Following your suggestion, we have revised this issue and checked the entire article carefully.
Reviewer Comment P 2.5—Some recent works about Image Classification Algorithms should be added such that: https://doi.org/10.32604/cmc.2022.019790.
Response: We thank the reviewer for this pertinent suggestion. Following your suggestion, we have cited this article and added a paragraph related to it as follows:
2.4. Deep Learning Models [39]
The ResNet-50 architecture was used as the CNN in this study. Many studies have shown that the fineness of detail that can be extracted by a CNN increases with the net-work depth. However, He et al. (2016) demonstrated that a degradation occurs if the depth increases beyond a certain point [39]. A residual network (ResNet) is an architecture based on residual blocks, which consists of convolution, activation, and batch normalization (BN) layers, F(x), and a shortcut connection that reproduces the input, x. Because the out-put of a residual block is H(x) = F(x) + x, the layers in a traditional network effectively learn the difference between the true output and x, i.e., the residual, as shown in Fig. 2. Therefore, if the network has not learned any features and the input is already optimal, F(x) is ap-proximately 0, or simply, H(x) = x (i.e., the identity relation). This solves the degradation problem and allows for extremely deep networks. The ResNet architecture is shown in Fig. 3. By employing deep learning models for image classification, it is possible for computers to automatically classify and label images for a variety of applications [40].
Reviewer Comment P 2.6—Mention the future scope or your plan to extend this work.
Response: We thank the reviewer for this valuable comment. We have addressed this issue in the Conclusions section as follows:
In the future, the proposed ResNet model could be combined with object detection or image segmentation strategies, such as SSD, Inception, or U-Net, to effectively detect tiny kidney stones in images. In addition, image classification is just the first step of our plan. Once the classification model is complete, we shall study object detection and segmentation. This is because our goal is to locate and label all kidney stones in a KUB image, and each image may contain one or many objects of varying types. For object detection, we expect to employ RetinaNet [56], which adds the single-shot multibox detector (SSD) to the frontend of ResNet and utilizes a focal loss function to improve image classification ac-curacy on unbalanced data (which is often the case for medical data). However, object detection will only provide rectangles that enclose a feature, not the exact profile of the object, which can be crucial for diagnosing a condition. Therefore, image segmentation is a quintessential part of AI-driven CAD. To this end, we expect to use CaraNet as our image seg-mentation model [57]. In each 1000 × 1000 px KUB image, a kidney stone may only occupy a region smaller than 20 × 20 px. Because CaraNet is specifically made for the segmentation of small objects, one of our future goals is to study the feasibility of using CaraNet to improve the segmentation of small kidney stones in KUB images
Reviewer Comment P 2.7—The authors must show how the effectiveness of the used model be measured.
Response: We thank the reviewer for this pertinent suggestion. Following your suggestion, we have given a more detailed explain about the performance evaluation measures of section 3.3 Experimental Results as follows:
The model was trained using 50 epochs, with an initial learning rate of 10-5. Because appropriate decreases to the learning rate are conducive for optimization, the learning rate was multiplied by 0.5 if the validation loss was not updated for 5 continuous epochs. The epoch-wise changes in accuracy and loss are shown in Figs. 8(a) and 8(b), respectively. According to the confusion matrix shown in Table 2, the final accuracy of the model was 0.977, and its accuracy on the testing set was 0.982. Sensitivity is the ratio of kidney stone patients who were correctly identified as positive cases, while precision is the ratio of cor-rect diagnoses among positive cases. Therefore, a high sensitivity implies that false nega-tives are rare. Specificity is the ratio of patients without a kidney stone who were correctly diagnosed as negative cases. Therefore, a model with a high specificity is unlikely to mis-diagnose healthy people as positive cases. The F1-measure is the harmonic mean of recall (sensitivity) and precision, which summarizes the performance of a model. In kidney stone classification, the focus is on sensitivity, as the primary goal is to correctly identify patients who suffer from kidney stones. The sensitivity, specificity, precision, and F1-measure scores of our model were 0.953, 1, 1, and 0.976 on the validation set and 0.964, 1, 1, and 0.982 on the testing set, respectively.
Reviewer Comment P 2.8—Check typos and grammar errors with some standard tool, it seems before submission revision, it is necessary to scan, and it will good English checker.
Response: We thank the reviewer for this pertinent suggestion. Following your suggestion, the manuscript has been improved by employing the English language editing services of a professional editor.
Reviewer Comment P 2.9—The contribution of the paper can be neatly elaborated by including the novelty of the paper and the major difference from available models.
Response: We thank the reviewer for this pertinent suggestion. Following your suggestion, we have addressed this issue through the following sentences in the Abstract section as follows.
The kidney ureter bladder (KUB) image is a radiological examination with low-cost, low-radiation and convenience. It is easy for clinicians in the emergency room to arrange KUB images as first-line examination for the patients with suspicious urolithiasis. However, it is hard to interpret the KUB images correctly by inexperienced clinicians. Also, it is difficult to get a formal report by the radiologist immediately after KUB image examination. Recently, artificial intelligence-based computer aided diagnosis (CAD) can effectively help non-expert clinicians to quickly make a correct diagnosis for further treatment. Therefore, the purpose of this study is to design a deep learning model for CAD of KUB images to help first-line clinicians in the emergency room make accurate diagnosis of urolithiasis. In this study, a total of 355 KUB images were retrospectively collected from 104 patients in Kaohsiung Chang Gung Memorial Hospital who were diagnosed with stones in their upper urinary tract. Then a deep learning model for potentially improving the accuracy in classifying KUB images in terms of the presence or absence of kidney stones was trained using ResNet on pre-processed KUB images and the parameters were tuned and tested. The experimental results indicate that the accuracy, sensitivity, specificity, and F1-measure of the model were 0.977, 0.953, 1, and 0.976 on the validation set, and 0.982, 0.964, 1, and 0.982 on the testing set, in addition, the results also demonstrate that the proposed model competes well against existing CNN-based model. In conclusion, we used a deep learning model to establish a computer-aided diagnosis for urolithiasis detection in Kidney Ureter Bladder images successfully. The model helps clinicians in the emergency room make accurate diagnosis and reduce unnecessary computed tomography (CT) related radiation exposure and waste of medical cost.

Reviewer 3 Report
The paper is well done and described. – In certain days, renal calculus has become a significant problem and if not detected at an early stage, then it's going to cause difficulties and sometimes surgery is additionally needed to get rid of the stone.
The article lacks data on the presence of ethical problems.
Author Response
Reviewer 3
Reviewer Comment P 3.1—The paper is well done and described. – In certain days, renal calculus has become a significant problem and if not detected at an early stage, then it's going to cause difficulties and sometimes surgery is additionally needed to get rid of the stone.
The article lacks data on the presence of ethical problems.
Response: We thank the reviewer for this pertinent suggestion. Following your suggestion, we obtained the IRB permission document, shown as follows. In addition, we sincerely thank you again for your positive response to this article.

Reviewer 4 Report
The authors have presented an exciting topic on designing and implementing a kidney stone detection model. The manuscript needs significant revisions a few of the important details are missing and a few aspects are too clearly presented to the reader
1. The title should highlight the technique used and should be elaborate for the reader to understand the method used.
2. Abstract needs significant improvement. Kindly organize the unstructured abstract consisting of the following information in order. Introduction to the topic, Aim of the study, Methods explored, significant results obtained, and outcomes with the overall conclusion.
3. Therefore, it may be surmised that an X-ray plain film is a cost-effective 40 alternative to CT, which also causes less harm to the human body. - What do you mean b this sentence?
4. Introduction: The content in the introduction looks slightly irrelevant content at the beginning. The authors can briefly talk about the technology and then about the previous related works. It needs major revision
5. Figure 1. Flowchart of the study outline - What is the purpose of this flowchart? It can be study-specific and then also mention the data used for testing, training, and validation as n =?
6. Figure 2. Residual Block[37] the citation can be provided in the manuscript text,
7. Figure 4. Histogram Equalization of KUB Image They should be numbered and the caption should be elaborated. at present from the figure, it is hard to depict anything as the clarity is poor of the histogram and the observations from the figures should be detailed in the manuscript text
8. In KUB images, histogram equalization (HE) is easily affected by the spine and pelvis areas. Why?
9. The authors should discuss in the detail the possible reason for the outcomes shown Training accuracy of ResNet model or any other models with significance rather than just quoting the values. What may be the technicality of the model developed which helps the performance to be better should be explained.
10. Discussion section can be further strengthened. There are several attempts by researchers on the topic. Their outcomes with values should be compared. Kindly consider recent at least 10-12 studies. Along with it a table showing a comparison of previous studies with the proposed method can be included.
11. The manuscript overall needs English language corrections and spell check
Best wishes to the authors for kindly revising the manuscript and presenting a good attempt for the consideration of publication
Author Response
Reviewer 4
Reviewer Comment P 4.1—1. The title should highlight the technique used and should be elaborate for the reader to understand the method used.
Response: We thank the reviewer for this valuable comment. Following your suggestion, we have retitled the paper as “Computer-aided Diagnosis for Urolithiasis Detection in Kidney Ureter Bladder Images using Deep Learning Model”.
Reviewer Comment P 4.2—Abstract needs significant improvement. Kindly organize the unstructured abstract consisting of the following information in order. Introduction to the topic, Aim of the study, Methods explored, significant results obtained, and outcomes with the overall conclusion
Response: We thank the reviewer for this pertinent suggestion. Following your suggestion, we have reorganized the Abstract as follows:
The kidney ureter bladder (KUB) image is a radiological examination with low-cost, low-radiation and convenience. It is easy for clinicians in the emergency room to arrange KUB images as first-line examination for the patients with suspicious urolithiasis. However, it is hard to interpret the KUB images correctly by inexperienced clinicians. Also, it is difficult to get a formal report by the radiologist immediately after KUB image examination. Recently, artificial intelligence-based computer aided diagnosis (CAD) can effectively help non-expert clinicians to quickly make a correct diagnosis for further treatment. Therefore, the purpose of this study is to design a deep learning model for CAD of KUB images to help first-line clinicians in the emergency room make accurate diagnosis of urolithiasis. In this study, a total of 355 KUB images were retrospectively collected from 104 patients in Kaohsiung Chang Gung Memorial Hospital who were diagnosed with stones in their upper urinary tract. Then a deep learning model for potentially improving the accuracy in classifying KUB images in terms of the presence or absence of kidney stones was trained using ResNet on pre-processed KUB images and the parameters were tuned and tested. The experimental results indicate that the accuracy, sensitivity, specificity, and F1-measure of the model were 0.977, 0.953, 1, and 0.976 on the validation set, and 0.982, 0.964, 1, and 0.982 on the testing set, in addition, the results also demonstrate that the proposed model competes well against existing CNN-based model. In conclusion, we used a deep learning model to establish a computer-aided diagnosis for urolithiasis detection in Kidney Ureter Bladder images successfully. The model helps clinicians in the emergency room make accurate diagnosis and reduce unnecessary computed tomography (CT) related radiation exposure and waste of medical cost.
Reviewer Comment P 4.3—Therefore, it may be surmised that an X-ray plain film is a cost-effective 40 alternative to CT, which also causes less harm to the human body. - What do you mean b this sentence?.
Response: We thank the reviewer for this pertinent suggestion. Let us explain our reasoning as follows:
In the absence of this report and sufficient experience, the diagnostician is likely to make mistakes or call for a CT, which delays treatment and increases treatment costs. To address this problem, we constructed a KUB-based deep learning CAD model to help frontline emergency physicians diagnose kidney stones and refer patients to urologists.
Reviewer Comment P 4.4—Introduction: The content in the introduction looks slightly irrelevant content at the beginning. The authors can briefly talk about the technology and then about the previous related works. It needs major revision.
Response: We thank the reviewer for valuable suggestion. Following your suggestion, we have re-organized the Introduction section and add a paragraph as follows:
Generally, experienced medical experts are needed to diagnose medical conditions from medical images. The manual effort this requires often makes it difficult to obtain timely diagnoses. Therefore, AI-based computer aided diagnosis (CAD) could be used by non-experts to rapidly obtain an accurate diagnosis for subsequent treatment. It is well-known that kidney, ureter, and urinary bladder (KUB) X-ray images only require low radiation doses and are relatively easy to obtain whilst being cost efficient. In addition, KUB imaging can be arranged at any time as a first-line imaging modality for kidney stones. However, this approach has poor diagnostic rates, as a very experienced urologist or radiologist is required to diagnose kidney stones from KUB images. Furthermore, the emergency physician who first saw the patient will not be able to immediately prepare a formal report for the diagnostician. In the absence of this report and sufficient experience, the diagnostician is likely to make mistakes or call for a CT, which delays treatment and increases treatment costs. To address this problem, we constructed a KUB-based deep learning CAD model to help frontline emergency physicians diagnose kidney stones and refer patients to urologists.
Reviewer Comment P 4.5—Figure 1. Flowchart of the study outline - What is the purpose of this flowchart? It can be study-specific and then also mention the data used for testing, training, and validation as n =?.
Response: We thank the reviewer for this pertinent suggestion. Following your suggestion, we have re-organized Figure 1.
Reviewer Comment P 4.6—Figure 2. Residual Block[37] the citation can be provided in the manuscript text,.
Response: We thank the reviewer for this pertinent suggestion. Following your suggestion, we have added this citation to the manuscript.
Reviewer Comment P 4.7—Figure 4. Histogram Equalization of KUB Image They should be numbered and the caption should be elaborated. at present from the figure, it is hard to depict anything as the clarity is poor of the histogram and the observations from the figures should be detailed in the manuscript text.
Response: We thank the reviewer for this pertinent suggestion. Following your suggestion, we have revised Figures 4–6 as follows:
Figure 4. Many areas are overexposed in the HE-processed image, especially around dense tissue, like bones.
Figure 5. Masking the spine and pelvis greatly decreases high-intensity areas.
Figure 6. Contrast limited adaptive histogram equalization (CLAHE) of a KUB image. It is clear that CLAHE greatly reduces overexposure around the rib cage, which makes it easy to observe kidney stones.
Reviewer Comment P 4.8—In KUB images, histogram equalization (HE) is easily affected by the spine and pelvis areas. Why?.
Response: We thank the reviewer for this pertinent suggestion. Following your suggestion, we have re-organized Section 3.1 Image Preprocessing with Histogram as follows:
When histogram equalization (HE) is performed on KUB images, overexposure often occurs at the spine and pelvis, which will negatively affect model training. In Fig. 4, it is shown that the HE-processed image is overexposed around dense tissues (bone), especially around the pelvis and spine, which may induce deviations during the feature ex-traction process. Therefore, it is necessary to first mask all high-density regions in the KUB images. As shown in Fig. 5, masking the spine and pelvis greatly reduces the high-intensity area. Nonetheless, some overexposure still occurs at the rib cage, which is a common problem in HE. To prevent image overexposure from HE, CLAHE was per-formed on the KUB images. In Fig. 6, it is clear that the CLAHE-processed image has little overexposure. Therefore, the CLAHE-processed KUB images are suitable for the observation of kidney stones. In addition, we are sorry for the misunderstanding caused by the discussion. The equalization itself will not be affected by the spine and pelvic area, but in the subsequent training process, high-density and highlighted areas such as the spine and pelvis may cause deviations in feature extraction. , so our primary goal is to remove the high-density areas in the plain x-ray film, and when performing histogram equalization, these high-density areas will also cause the image to be overexposed.
Reviewer Comment P 4.9—The authors should discuss in the detail the possible reason for the outcomes shown Training accuracy of ResNet model or any other models with significance rather than just quoting the values. What may be the technicality of the model developed which helps the performance to be better should be explained..
Response: We thank the reviewer for this pertinent suggestion. Following your suggestion, we have given a more detailed explain about the performance evaluation measures of section 3.3 Experimental Results as follows:
The model was trained using 50 epochs, with an initial learning rate of 10-5. Because appropriate decreases to the learning rate are conducive for optimization, the learning rate was multiplied by 0.5 if the validation loss was not updated for 5 continuous epochs. The epoch-wise changes in accuracy and loss are shown in Figs. 8(a) and 8(b), respectively. According to the confusion matrix shown in Table 2, the final accuracy of the model was 0.977, and its accuracy on the testing set was 0.982. Sensitivity is the ratio of kidney stone patients who were correctly identified as positive cases, while precision is the ratio of cor-rect diagnoses among positive cases. Therefore, a high sensitivity implies that false nega-tives are rare. Specificity is the ratio of patients without a kidney stone who were correctly diagnosed as negative cases. Therefore, a model with a high specificity is unlikely to mis-diagnose healthy people as positive cases. The F1-measure is the harmonic mean of recall (sensitivity) and precision, which summarizes the performance of a model. In kidney stone classification, the focus is on sensitivity, as the primary goal is to correctly identify patients who suffer from kidney stones. The sensitivity, specificity, precision, and F1-measure scores of our model were 0.953, 1, 1, and 0.976 on the validation set and 0.964, 1, 1, and 0.982 on the testing set, respectively.
Reviewer Comment P 4.10—Discussion section can be further strengthened. There are several attempts by researchers on the topic. Their outcomes with values should be compared. Kindly consider recent at least 10-12 studies. Along with it a table showing a comparison of previous studies with the proposed method can be included.
Response: We thank the reviewer for this pertinent suggestion. Following your suggestion, we have revised the section 4. Discussion section as follows:
In this study, a CNN was used to train a model that can classify KUB images ac-cording to the presence of kidney stones. Although few studies have been conducted on the use of X-ray plain films for the detection of kidney stones, the results of these studies are promising. Reference [47], which is a review of recent AI advancements in urology, only mentioned one study that was based on KUB images [27]. The remainder were ma-chine and deep learning models based on CT images, like in Reference [48].
Reviewer Comment P 4.11—The manuscript overall needs English language corrections and spell check.
Response:
We thank the reviewer for this pertinent suggestion. Following your suggestion, the manuscript’s language has been improved by employing the English language editing services of a professional editor.
Reviewer Comment P 4.12—Best wishes to the authors for kindly revising the manuscript and presenting a good attempt for the consideration of publication.
Response:
Thank you very much for your valuable comments on this article. We will continue to work hard, and we have carefully and sincerely addressed all comments and suggestions in the revised manuscript.

Round 2
Reviewer 2 Report
The authors answered all comments
Author Response
Reviewer 2
Reviewer Comment P 2.1—The authors answered all comments
Response: Thank you very much for your valuable comments on this article. We will continue to work hard, and we have carefully and sincerely addressed all comments and suggestions in the revised manuscript.

Reviewer 4 Report
The authors have presented the manuscript well. The major changes suggested are incorporated. However, the recent studies on a similar topic are missing out
https://doi.org/10.1089/end.2020.1136
https://doi.org/10.3390/jcm11175151
2. , like in Reference [48] and Reference [47] the authors need to revise by mentioning as discussed by Author name et al. [47]. Kindly revise the manuscript.
3. Kindly recheck if all the citations in the text of the manuscript are listed in the references.
4. What is the need for Table 1. It can be included as a figure with the mention of the dataset with respect to the study and for the test, training, and validation.
5. Inferences from Fig 8 and Fig 9 should be discussed. At present only mention the presence of figures in mentioned in the manuscript text.
6. Is there any specific reason why authors have compared the proposed method in the study with only CNN-based model [27]? It would be better if it is compared and discussed with at least 5-6 similar studies.
7. Language and Spell corrections and few instances in the manuscript need to be rechecked
Author Response
Reviewer 4
Reviewer Comment P 4.1—The authors have presented the manuscript well. The major changes suggested are incorporated. However, the recent studies on a similar topic are missing out https://doi.org/10.1089/end.2020.1136 https://doi.org/10.3390/jcm11175151 Response: Thank you for this pertinent suggestion. Following your suggestion, we have cited this article and added a paragraph related to it as follows. Reviewer Comment P 4.2—like in Reference [48] and Reference [47] the authors need to revise by mentioning as discussed by Author name et al. [47]. Kindly revise the manuscript. Response: Thank you for this pertinent suggestion. Following your suggestion, we have changed the way the specific reference was mentioned in the manuscript. Reviewer Comment P 4.3—Kindly recheck if all the citations in the text of the manuscript are listed in the references. Response: Thank you for this pertinent suggestion. Following your suggestion, we have double-checked all citations in this revised manuscript. Reviewer Comment P 4.4—What is the need for Table 1. It can be included as a figure with the mention of the dataset with respect to the study and for the test, training, and validation. Response: Thank you for this pertinent suggestion. Following your suggestion, we have re-organized Figure 4 to explain the result of Table 1 of the revised manuscript. Reviewer Comment P 4.5—Inferences from Fig 8 and Fig 9 should be discussed. At present only mention the presence of figures in mentioned in the manuscript text. Response: Thank you very much for this observation. Following your suggestion, we have revised Section 3.3. Experimental Results as follows. The model was trained for 50 epochs with an initial learning rate of 10-5. Because appropriate decreases to the learning rate are conducive for optimization, the learning rate was multiplied by 0.5 if the validation loss was not updated for 5 continuous epochs. The epoch-wise changes in accuracy and loss are shown in Figs. 9(a) and 9(b), respectively. It may be observed that the process of training from 0 epochs to 20 epochs converged rapidly. The accuracy and loss of the training set and the verification set were close, indicating that the model learned features in the initial stage well and classified them accurately. The subsequent loss from the 20th to the 50th epochs gradually converged to the optimal solution as the training ended. According to the confusion matrix shown in Table 1, the final accuracy of the model was 0.977, and its accuracy on the testing set was 0.982. Sensitivity is the ratio of patients with kidney stones who were correctly identified as positive cases, while precision is the ratio of correct diagnoses among positive cases. Therefore, a high sensitivity implies that false negatives are rare. Specificity is the ratio of patients without kidney stones who were correctly diagnosed as negative cases. Therefore, a model with a high specificity is unlikely to misdiagnose healthy subjects as positive cases. The F1-measure is the harmonic mean of recall (sensitivity) and precision, which summarizes the performance of a model. In kidney stone classification, the focus is on sensitivity, as the primary goal is to correctly identify patients who suffer from kidney stones. The sensitivity, specificity, precision, and F1-measure scores of our model were 0.953, 1, 1, and 0.976 on the validation set and 0.964, 1, 1, and 0.982 on the testing set, respectively (see Table 2). The ROC curves were also plotted to test the effectiveness of the model, and their AUCs were 0.995 and 1 on the validation and testing sets, respectively (Figs. 10(a) and 10(b)). When AUC > 0.5, the classification performance of a classifier is better than random guessing, and the model has positive predictive value. The AUC value of our model was quite close to 1, which shows that the performance of our model was close to that of a theoretically perfect classifier, and it was effective in predicting positive samples correctly. Reviewer Comment P 4.6—Is there any specific reason why authors have compared the proposed method in the study with only CNN-based model [27]? It would be better if it is compared and discussed with at least 5-6 similar studies. Response: Thank you for this pertinent suggestion. Following your suggestion, we have revised Section 4. Discussion as follows. In this study, we trained a CNN model to classify KUB images according to the presence of kidney stones. Although few studies have been conducted on the use of plain film X-ray images to detect kidney stones, the results are promising. According to a recent systematic review of the recent AI advancements in urology by Dai et al. [49], only a single study used KUB images [29]. Other studies largely considered machine and deep learning models based on CT images, such as a work by Parakh et al. [50]. Reviewer Comment P 4.7—Language and Spell corrections and few instances in the manuscript need to be rechecked Response: Thank you for this pertinent suggestion. Following your suggestion, the quality of the language used in the manuscript has been further improved and the entire text has been double-checked by an English language editing service.
